# Kidney and Kidney Tumor Segmentation using Spatial and Channel attention enhanced U-Net

Sajan Gohil[1*], Abhi Lad[1*]

[1]Pandit Deendayal Energy University, Gandhinagar, India
sajan.gict16@sot.pdpu.ac.in
abhi.lce16@sot.pdpu.ac.in
*Equal Contribution

**Abstract.** Kidney and Kidney tumor segmentation from CT scans has tremendous potential to help doctors in early diagnosis and localization of tumor, its size and type and for making timely treatment plans. However, considering the nature and volume of data, it is difficult and time consuming to train on such CT scans. In this paper, we propose enhancements to the 3D U-Net model to incorporate Spatial and Channel Attention in order to improve the identification and localization of segmentation structures by learning on spatial context. When compared with Residual U-Net model with greater depth and more feature maps, our Spatial and Channel Attention enhanced U-Net with less depth and feature maps performed significantly better on validation and training set when trained under similar conditions.

**Keywords:** Kidney Tumor, segmentation, 3D U-Net, Spatial Attention, Channel Attention

## 1 Introduction

Manually identifying tumors from CT scans is a tedious and time-consuming process. It is also a difficult task as there can be inconsistencies in proper segmentation even by experienced practitioners. In some cases, the boundaries of lesions can also be unclear in CT scan images and the images can also have poor contrast and structure definition. To help solve these issues, many computer vision based deep learning methods have been proposed and developed which are trained to segment and/or classify such lesions. One of the most popular models for such tasks is U-Net [1] which can be modified for 3D convolutions and with residual units as proposed in [2]. Furthermore, the size of such imaging modalities can be huge, thus making it difficult to train on all the data. So proper data preprocessing and manipulation can also play an important role in making the training efficient and viable and make the model more robust to drift in data.

In this paper, we enhance the basic U-Net by including visual attention introduced in [3] with 3D convolutions. Attention blocks help our model train more effectively by refining the features with the help of a global attention map. Combination of attention mechanism with U-Net has previously been explored in [4]. [5] introduced Squeeze and excite mechanism which is advantageous for channel refinement as used in works such as [6] with variations of attention mechanism across spatial and channel dimensions and [7] with project and excite blocks for segmentation of volumetric medical scans. Models with attention have previously been used for Kits19 [8] dataset in works such as [9] which use attention modules at the end of their model for final refinement of features. In this paper, we use architecture is similar to [10] which uses spatial and channel attention separately for fine and sparse features. We also trained a standard U-Net with residual blocks for comparison and found that our spatial and channel attention enhanced U-Net performed better on training and validation sets, that too with less number of epochs. We also resample our data, especially along the z-axis to effectively increase the number of training image slices per volume. In order to reduce training time on constrained resources, we use Nvidia Clara SmartCache [11] to improve training times without loading the entire dataset to cpu memory. The following sections describe in detail the data methods and the custom deep learning U-Net model used in our challenge submission.

## 2 Methods

In this section we discuss the data split for training, preprocessing steps and data augmentations from the point of view of generalizability, and finally the custom U-Net model along with the details about attention blocks.

## 2.1 Training and Validation Data

Our submission made use of the official KiTS21 training set alone. For segmentation masks, we have used "aggregated_AND" based final masks to train our model. We have refrained from using data from other similar studies and instead use data augmentation techniques to adapt the segmentation model for better generalizability. The data was split in a 90:10 ratio for training and validation respectively. The data is randomly shuffled before creating the validation split. The final submission uses a model trained on the entire training set.

## 2.2 Preprocessing

The number of slices for each volume sample is different and the number of slices can also be high enough to consume all available resources during training. To avoid such a scenario, we resample the data to 2x1.62x1.62 mm to have the same voxel spacing across the patient image volumes. The lower spacing along the z-axis increases the number of training slices per patient and thus helps generalizability. We further clean the data by keeping only the body structures in frame by cropping the foreground. The final cropped data is further divided in chunks of 64x128x128 volumes for training and evaluation. For training Residual U-Net, we divide the cropped data in chunks of 64x160x160 to capture higher spatial information.

The intensity values vary based on physical properties of structures and thus we remove unnecessary values corresponding to structures like bone and air by clipping to range (-80, 305). Then the clipped values are rescaled between 0 and 1 for all images.

In order to improve training times and prevent crashes by cpu memory bottleneck, we use the Nvidia Clara SmartCache dataset loader. We set the cache rate = 0.4 and replace rate = 0.5. This ensures sufficient data is cached in memory for training while replacing 50% of cached data at each step without caching the entire dataset and choking the cpu memory.

## 2.3 Data Augmentations

In order to make the model generalizable, we introduce variance in data using data augmentation techniques. To introduce spatial variance, we use 3D Elastic deformation with a probability of 0.5. The parameters for 3D elastic deformation are: sigma (smoothness factor) = (5,8); magnitude = (50,150); translate = (10,10,5) in pixels; rotate = (5,5,180) in degrees, scale = (0.1, 0.1, 0.1) in proportion of image size. This augmentation introduces variation in shape of structures while maintaining spatial information.

For introducing perceptual or intensity variations, we use random intensity shift with maximum intensity offset value of 0.1. We also use Gaussian noise with mean = 0 and standard deviation = 0.1. Both of the intensity-based augmentations are applied with probability of 0.25 individually.

## 2.4 Proposed Method

Following the success of U-Net models and its variants, we have decided to use the U-Net model with 3D Convolution blocks as our base architecture. Our base architecture has 3 encoding and 3 decoding blocks with a bottleneck block in between. Fig. 1 (A) describes the architecture used in our submission. The number of feature maps for encoder blocks are (32,64,128), followed by 128 feature maps for the bottleneck block. The feature maps from skip connections are stacked in decoding blocks resulting in feature maps (256, 128, 64). All the encoding and decoding blocks have kernel size of 3x3x3 and stride value as 2, except for bottleneck block with kernel size of 3x3x3 and stride value 1, and the final output conv layer with kernel size 1x1x1 and stride value 1. To enhance this base model, we have added Spatial attention and Channel attention modules, as introduced in [12] in an architecture similar to [10] which was proposed for 2D segmentation tasks.

*Spatial Attention block:* The spatial attention block is responsible for identifying where the useful information is present in the image, by utilizing the inter-spatial relationships of the image features. Fig. 1 (B) describes the spatial

attention block used in our proposed approach. The spatial attention block uses mean and max operations along channel dimension followed by 3D conv (7x7x7) to identify region of interest and multiply the attention map with output of preceding 3D Conv + ADN (Attention+Dropout+Norm) block to filter out location of important features.

*Channel Attention block:* Instead of focusing on where the important feature is, the channel attention block identifies what is useful in a given image. Fig. 1 (C) describes the channel attention block used in our approach. The channel attention block uses mean and max values across spatial dimensions followed by a conv block to identify what is important in a given volume.

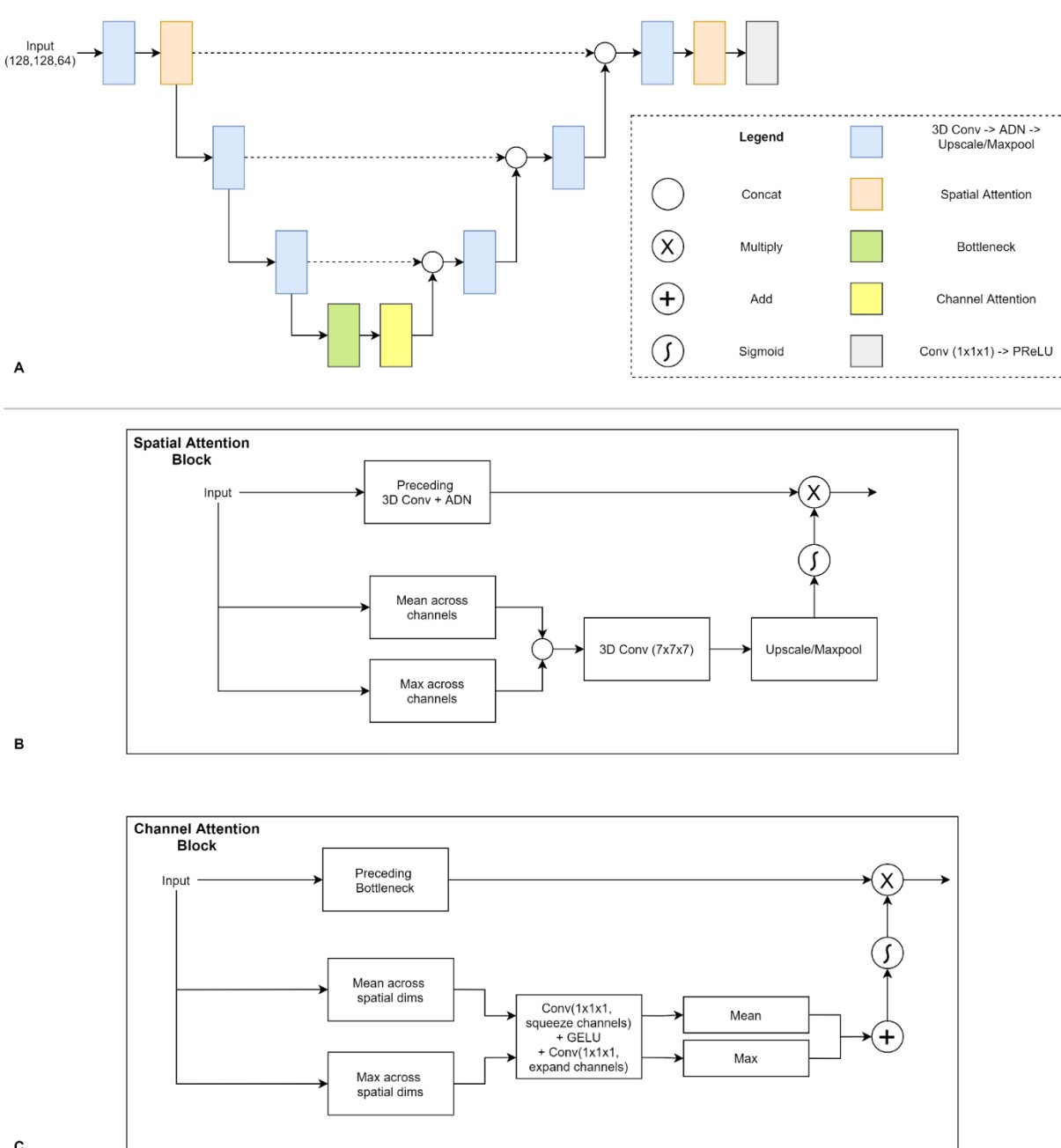

**Fig. 1**: (A) describes the enhanced U-Net architecture used in our submission. (B) represents the working of Spatial Attention Block. (C) represents the working of Channel Attention Block. (B) and (C) includes the interaction of attention mechanism with preceding blocks in the U-Net.

Our implementation is similar to model architecture of [10]. We have added Spatial attention to the first encoding and last decoding blocks having the largest dimensions. We have added channel attention to the bottleneck layer. The depth and number of feature maps of our enhanced U-Net has been limited due to resource constraints but can be increased to further improve the results.

### 2.5 Residual U-Net for Comparison

We also train a residual 3D U-Net to compare performance of our proposed model. The residual U-Net is similar to [13], however the number of features maps are different (16,32,64,128,256). The absence of attention blocks allows for a greater number of encoding-decoding blocks with more number of feature maps. Also, as compared to 64x128x128 patch size used for training our proposed model, we increase the size of patches to 64x160x160 for residual U-Net to increase the spatial information per patch

### 2.6 Implementation and Training

The model is implemented in pytorch using MONAI [11] framework. Both the models output 4 channels corresponding to 4 classes including the background. For training we have used DiceCE loss which is a combination of Dice and Cross Entropy loss functions. We have used AdamW optimizer with a learning rate of 10e-4. Both models are trained for 300 epochs for comparison and the final submission is made using our proposed model which is trained for 500 epochs on Nvidia P100 gpu with 16GB vram and 24GB cpu ram. Validation is performed at the end of each epoch using the overall dice score as metric. Each epoch takes ~1.5 minutes and the whole model completes 500 epochs in ~13 hours.

### 2.7 Inference Procedure

For final inference, since the model is trained on 64x128x128 sized chunks of input volume, we use sliding window inference with overlap = 0.8. The high overlap value increases the inference time but also improves the segmentation results. The 4 channeled output is converted to 1 channel by using maximum probability of segmented classes. And finally, the voxel spacing of the segmented volumes are restored to original spacing of the input volume by inverse transform.

## 3  Results

Here we provide the comparative results of our proposed model with residual U-Net based on training and validation set. We provide overall dice and structure specific dice scores. Finally, we also provide the dice scores on the test set as provided by kits21 evaluation system. Table I shows the performance of our proposed model and Residual U-Net on validation and complete training set after 300 epochs. Our approach outperformed Residual U-Net by margin or ~0.2 in mean Dice score on both validation and training sets.

**Table I:** Dice score for individual classes and mean across classes of our approach compared with Residual U-Net on validation and complete training set after 300 epochs.

|  | Set | Kidney | Tumor | Cyst | Mean |
|---|---|---|---|---|---|
| **Residual U-Net** | Validation Set | 0.901 | 0.456 | 0.273 | 0.543 |
| **Our approach** |  | **0.952** | **0.665** | **0.656** | **0.757** |
| **Residual U-Net** | Complete Training Set | 0.901 | 0.480 | 0.270 | 0.550 |
| **Our approach** |  | **0.937** | **0.683** | **0.523** | **0.714** |

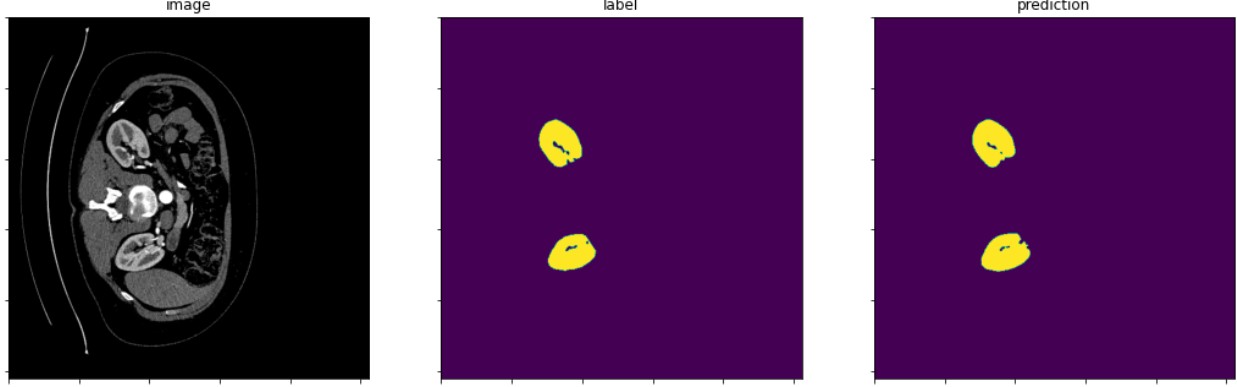

**Fig. 2:** Visualization of predicted segmentation map and corresponding label for an image from validation set.

Fig. 2 shows the segmentation results corresponding to a volume from validation set of our proposed model after 300 epochs. As it can be seen, the results do not suffer from jagged/rough edges from resampling the input volume.

The results on the test set of KiTS21 are summarized in Table II.

**Table II:** Dice score for individual classes and mean across classes of our approach on test set.

|  | Kidney | Tumor | Cyst | Mean |
|---|---|---|---|---|
| **Our approach** | Awaiting |  |  |  |

## 4  Conclusion

In this paper, we propose an enhancement for existing 3D U-Net model using attention-based blocks. The model we have used is a modified version of U-Net with spatial and channel attention modules. We preprocess and augment the data to improve the generalizability of our segmentation model. We compare our architecture (Mean Dice: 0.757) with Residual U-Net (Mean Dice: 0.543) architecture and show that our proposed architecture, which is inferior in depth and number of feature maps as compared to Residual U-Net, manages to outperform the Residual U-Net by significant margin on validation set after being trained for similar number of epochs. Our proposed approach has potential to improve results by further increasing the depth and number of feature maps and using a larger sized chunks to improve spatial context.

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
