# OpenReview forum: "Kidney and Kidney Tumor Segmentation using Spatial and Channel attention enhanced U-Net"
_MICCAI.org/2021/Challenge/KiTS — Submitted to KiTS21 Challenge_

### Official Review · Reviewer_sE6Q · 2021-08-30

**Rating:** 5

**Review:**

This paper presents a u-net based approach to segmentation that focuses most heavily on unconventional preprocessing strategies that allow for training on systems with relatively few resources. The paper is fairly short and could benefit from expansion in just about every section. The results are currently left as a placeholder but the authors should be sure to expand beyond just the single sentence once the true results are known. One or two figures would also go a long way towards helping the submission.

One crucial detail that was left out was how the multiple annotations per case were synthesized together during training. Did you use majority voting? Please make sure to add this detail.

---

### Official Review · Reviewer_vyKa · 2021-08-30

**Rating:** 5

**Review:**

### Overall

- Please rephrase "on a resource system" in the abstract - is there a word missing?

### Introduction

- It would be nice to end this section with a brief overview of how you intended to approach this problem.

### Methods

- When you say that you added non continuous slices - they retained their order though, right? Please clarify within the paper.
- Please expand on your augmentation strategy. Could you provide some paramters for the transformations?
- It would be nice if you could add a figure to this section to help the reader to understand your approach. A simple flow diagram or representation of your architecture would suffice

### Results

- Please expand on your validation results. Did you test any other approach? How did it compare to the approach that you submitted?
- It would be great if you could include a figure here that shows your prediction and the corresponding ground truth.
- Please be sure to add the official results once they are known


### Discussion and Conclusion

- It would be nice to also summarize your performance here

---

### Decision · Program_Chairs · 2021-08-30

**Decision:**

Major Revisions

**Comment:**

Please address the reviewer comments and resubmit